

# Core proteome mediated subtractive approach for the identification of potential therapeutic drug target against the honeybee pathogen *Paenibacillus larvae*

Sawsen Rebhi[1,*], Zarrin Basharat[2,*], Calvin R. Wei[3], Salim Lebbal[4], Hanen Najjaa[5], Najla Sadfi-Zouaoui[1] and Abdelmonaem Messaoudi[1,6]

[1] Université de Tunis-El Manar, Laboratoire de Mycologie, Pathologies et Biomarqueurs, Département de Biologie, Tunis, Tunisia

[2] Independent researcher, Islamabad, Pakistan

[3] Department of Research and Development, Shing Huei Group, Taipei, Taiwan

[4] University of Khenchela, Department of Agricultural Sciences, Faculty of Nature and Life Sciences, Khenchela, Algeria

[5] University of Gabes, Laboratory of Pastoral Ecosystem and Valorization of Spontaneous Plants and Associated Microorganisms, Institute of Arid Lands of Medenine, Medenine, Tunisia

[6] Jendouba University, Higher Institute of Biotechnology of Beja, Beja, Tunisia

[*] These authors contributed equally to this work.

## ABSTRACT

**Background & Objectives**. American foulbrood (AFB), caused by the highly virulent, spore-forming bacterium *Paenibacillus larvae*, poses a significant threat to honey bee brood. The widespread use of antibiotics not only fails to effectively combat the disease but also raises concerns regarding honey safety. The current computational study was attempted to identify a novel therapeutic drug target against *P. larvae*, a causative agent of American foulbrood disease in honey bee.

**Methods**. We investigated effective novel drug targets through a comprehensive *in silico* pan-proteome and hierarchal subtractive sequence analysis. In total, 14 strains of *P. larvae* genomes were used to identify core genes. Subsequently, the core proteome was systematically narrowed down to a single protein predicted as the potential drug target. Alphafold software was then employed to predict the 3D structure of the potential drug target. Structural docking was carried out between a library of phytochemicals derived from traditional Chinese flora ($n > 36,000$) and the potential receptor using Autodock tool 1.5.6. Finally, molecular dynamics (MD) simulation study was conducted using GROMACS to assess the stability of the best-docked ligand.

**Results**. Proteome mining led to the identification of Ketoacyl-ACP synthase III as a highly promising therapeutic target, making it a prime candidate for inhibitor screening. The subsequent virtual screening and MD simulation analyses further affirmed the selection of ZINC95910054 as a potent inhibitor, with the lowest binding energy. This finding presents significant promise in the battle against *P. larvae.*

**Conclusions**. Computer aided drug design provides a novel approach for managing American foulbrood in honey bee populations, potentially mitigating its detrimental effects on both bee colonies and the honey industry.

Corresponding author
Abdelmonaem Messaoudi,
messaoudiabdelmonemster@gmail.com

## INTRODUCTION

The majority of plant species crucial for our food supply rely on insect pollinators. Honey bees (scientific name: *Apis mellifera*) play a crucial role in global fruit cultivation, with over 90% of crops depending on their pollination services (*Zheng et al., 2018*). This vital service has a direct impact on human food consumption and sustains biodiversity by supporting the pollination of flowering plants and crops. Approximately 35% of the food consumed by people depends on bee-mediated pollination (*Klein et al., 2007*). Products such as beeswax and royal jelly, in addition to honey, are produced by bees and have been utilized for ages in food, medicine, and cosmetics (*Dumitru et al., 2022*).

Honeybees confront various disease-causing agents, including bacteria, viruses, protozoa, fungi, and parasitic mites (*Neov et al., 2019*). Notable among these pathogens is *Paenibacillus larvae*, the causative agent of American foulbrood disease (AFB), a highly destructive and contagious disease affecting *Apis mellifera* larvae (*Ye et al., 2023*). This Gram-positive, facultative anaerobic, spore-forming bacterium is capable of infecting honeybee brood within the first three days of their lives. It can also impact the pupal stage (*Daisley et al., 2023*). This pathogen has a global presence, with viable spores persisting in hives for extended periods, even decades, posing a continuous threat to honeybee colonies and their health. Given its virulence and significant impact on honey bee colonies, AFB is internationally classified as a notifiable disease, requiring mandatory reporting worldwide (*Dickel et al., 2022*). Considering the severity of the infection, a frequent occurrence, rapid and easy spread, epizooty and enzooty are common (*Ebeling et al., 2023*). The classification of AFB as a highly dangerous infectious animal disease by the World Organization for Animal Health underscores the severity and rapid spread of this disease within apiaries (*Matovic et al., 2023*).

When clinical signs of the disease become apparent in a hive, a common practice to control its spread is by resorting to extreme measures such as burning the hive, equipment, and the entire colony. Additionally, antibiotics like oxytetracycline hydrochloride (marketed as Terramycin) or tylosin (a macrolide) are sometimes employed to inhibit the proliferation of vegetative cells of *P. larvae* in the midgut of honeybee larvae. However, in some regions, the prophylactic use of these antibiotics has led to the emergence of antibiotic-resistant strains of *P. larvae* (*Alippi et al., 2007*; *Genersch et al., 2010*). Consequently, due to the risk of developing resistant strains and potential contamination of honeybee products, several European Union nations have prohibited the use of antibiotics in beekeeping practices.

It is essential to clarify that antibiotics are primarily utilized for antimicrobial metaphylaxis of AFB, aiming to prevent outbreaks rather than treating established clinical infections in hives affected by AFB. Furthermore, it's worth noting that there are no Maximum Residue Limits (MRLs) established for antibiotics in honey, according to

European Community laws (*Mutinelli, 2003*), which prohibits the sale of honey containing antibiotic residues. This highlights the need for alternative, sustainable approaches to manage *P. larvae*. Integrated pest management (IPM) principles advocate for a judicious combination of methods, promoting the health of honeybee colonies while minimizing the environmental impact. By integrating IPM into beekeeping practices, we can strive for more effective and environmentally conscious strategies in the management and prevention of AFB outbreaks. IPM offers a comprehensive and sustainable approach that considers various factors such as biological, cultural, and mechanical controls alongside chemical interventions (*Bava et al., 2023*). It emphasizes ecological balance and minimizes the impact on non-target organisms. Additionally, it involves the utilization of substances known for their anti-*P. larvae* properties, encompassing both synthetic compounds such as (Thio) ether, sulfone, ester derivatives (*Šamšulová et al., 2023*) and naturally derived agents like fatty acids (such as algal metabolite lauric acid) (*Lopes et al., 2016*), essential oils (*Ansari et al., 2016*), plant extracts (*Isidorov et al., 2018*), propolis (along with its constituents) (*Wilson et al., 2015*), as well as probiotic bacteria (*Truong et al., 2023*).

It also calls for investigation of newer approaches and techniques to find effective treatments based on novel therapeutic targets for managing AFB by inhibiting *P. larvae*. In this context, computer-aided drug design (CADD), a swift method for efficiently identifying viable therapeutic candidates, can help to reduce the usage of animal models in pharmacological research as well as help with the rational design of new and safe drug candidates (*Niazi & Mariam, 2023*). It can facilitate repositioning of already-marketed medications. This support to medicinal chemists and pharmacologists is instrumental throughout the drug development process (*Brogi et al., 2020*). Furthermore, the exploration of plant-derived components as potential agents against AFB aligns with the rich historical utilization of traditional herbal medicine. Plant-derived compounds may offer effective and safe alternatives for managing AFB (*Abdallah et al., 2023*), contributing to a broader understanding of sustainable and eco-friendly strategies in apiculture. Recently, the antibacterial activity of inflorescences and roots extracts of Cannabis (*Cannabis sativa* L) against *P. larvae* has been reported (*Giselle et al., 2023*). Researchers have also reported tannin extract to be potent antimicrobials against *P. larvae* (*Giménez et al., 2021*). The extract of *Dicranum polysetum* has also shown a significant reduction in infection among honey bee larvae, with clinical cessation of infection upon extract administration within the initial 24 h after spore ingestion. The antimicrobial activity of these compounds does not compromise larval viability, live weight, or interact adversely with royal jelly and shows promise in addressing early-stage AFB infection without negative impacts on essential aspects of larval health. Further research is warranted to investigate the potential reduction of risk of developing resistance to these novel compounds (*Karaoğlu et al., 2023*).

This is the first subtractive proteomic study that aims to identify potential novel drug targets in *P. larvae*. Utilizing a target that has not been previously employed in antimicrobial therapy against a selected species, offers reduced risk of resistance development, as the bacteria may not have encountered selective pressures against these novel targets. The exploration of new targets also provides an opportunity for identifying compounds with unique mechanisms of action (*Aljeldah, 2022*). The chosen target was used to identify

potent inhibitors using natural product libraries by virtual screening, molecular docking and MD simulation. We hope that our research will contribute to the development of innovative drugsfor combating AFB and safeguarding honeybee populations.

## MATERIALS & METHODS

### Data retrieval and pan-genomic analysis

In the current study, available strains of *P. larvae* in NCBI (having complete genome) were considered for the pan-proteome analysis. A total of 14 such unique proteomes in .fasta format were retrieved from the NCBI database (*Kd PJNAR, 2007*) on April 11, 2022 (Table 1) and used as input files in the Bacterial Pan-genome Analysis (BPGA) (*Chaudhari, Gupta & Dutta, 2016*) pipeline.

The BPGA tool was utilized to conduct pan-proteomic analysis aimed at identifying the core proteome. Orthologous protein clusters were identified using the USEARCH tool (*Edgar, 2010*), employing a 70% cut-off for sequence identity. Specifically, proteins were categorized as follows: those present in all or nearly all strains constituted the core proteome, while proteins present in some but not all strains were classified as part of the accessory proteome. Proteins sporadically present across strains, lacking consistent presence, were designated as part of the cloud proteome. To generate a matrix reflecting protein presence or absence in each strain, the MUSCLE tool (*Edgar, 2004*) was employed with default settings for aligning the core, accessory, and unique proteins. The MUSCLE algorithm utilizes two distance measures, k-mer distance for unaligned pairs and Kimura distance for aligned pairs, enhancing speed and accuracy. The log-expectation (LE) score, for profile alignment, incorporating probabilities and frequencies derived from the modified point accepted mutation (PAM) matrix. The obtained input was used for inferring phylogenetic relationships between strains based on the neighbor-joining method. By default, MUSCLE supports Unweighted Pair Group Method with Arithmetic Mean (UPGMA) because it is computationally less intensive and often faster, making it suitable for large datasets (*Hua et al., 2017*). We selected neighbour joining (NJ) during tree construction option in BPGA (*Thorat et al., 2020*). NJ method does not strictly merge the closest neighbors at each step but evaluates the entire distance matrix to identify the pair that minimizes the total branch length (*Zhang & Sun, 2008*). It is less sensitive to violations of the molecular clock assumption and provides more reliable trees due to its consideration of varied evolutionary rates (*Holland, Penny & Hendy, 2003*). The pan- and core-proteome dot plots were generated and COG/KEGG orthologs were mapped to infer functional profile. COG categorizes proteins into orthologous groups based on their predicted functions (*Tatusov et al., 2000*). It helps in functional annotation by assigning similar functions to proteins that share common ancestors. KEGG provides a comprehensive resource for understanding biological pathways and systems. It includes information on metabolic pathways, regulatory pathways, and other cellular processes (*Okuda et al., 2008*).

### Subtractive proteomics analysis

The CD-HIT online server was used to identify all non-redundant proteins of the pathogen with a sequence similarity threshold of 60% (*Huang et al., 2010*; *Zoghlami et*

Rebhi et al. (2024), *PeerJ*, DOI 10.7717/peerj.17292

**Table 1  Statistics of *P. larvae* strains.** Information and analysis of *P. larvae* strains.

| | Project NCBI identifier | Name | Strain names | Status | No contigs | Genome size | % GC | No CDSs | No RNAr | No RNAs | No other RNA |
|---|---|---|---|---|---|---|---|---|---|---|---|
| 1 | PRJNA280999 | *P. larvae* | MEX14 | Complete | 2 | 4 .19 | 44.00 | 4,025 | 6 | 72 | 4 |
| 2 | PRJNA284352 | *P. larvae* | G25-75 | Complete | 264 | 4.56 | 44.10 | 4,519 | 26 | 79 | 4 |
| 3 | PRJNA32897 | *P. larvae* ssp. Larvae | ATCC 9545 | Complete | 1 | 4.29 | 44.20 | 3,942 | 24 | 80 | 4 |
| 4 | PRJNA35814 | *P. larvae ssp.* | ERIC-I | Complete | 1 | 4 .29 | 44.20 | 4,019 | 24 | 80 | 4 |
| 5 | PRJNA369467 | P. *larvae* ssp. Larves | Eric_V | Complete | 3 | 4.76 | 43.99 | 4,693 | 24 | 79 | 4 |
| 6 | PRJNA358155 | *P. larvae* ssp. | Eric_III | Complete | 4 | 4.70 | 43.98 | 4,468 | 24 | 79 | 4 |
| 7 | PRJNA369466 | *P. larvae* ssp. | Eric_IV | Complete | 3 | 4.38 | 44.19 | 4,133 | 24 | 79 | 4 |
| 8 | PRJNA46259 | *P. larvae* ssp B-3650 | B-3650 | Complete | 353 | 4.35 | 44.10 | 4,222 | 15 | 55 | 4 |
| 9 | PRJNA13476 | *P. larvae* ssp. BRL-230010 | BRL-230010 | Complete | 573 | 4.03 | 44.10 | 3,912 | 17 | 52 | 4 |
| 10 | PRJNA42205 | *P. larvae* ssp. DSM 25430 | DSM 25430 | Complete | 2 | 4.44 | 44.98 | 3,696 | 25 | 81 | 4 |
| 11 | PRJNA42203 | *P. larvae* ssp. DSM 25719 | DSM 25719 | Complete | 8 | 4.76 | 44.08 | 4,436 | 23 | 79 | 4 |
| 12 | PRJNA362897 | *P. larvae* ssp. *puvifaciens* | SAG 10367 | Complete | 2 | 4 .79 | 43.79 | 4,448 | 24 | 80 | 4 |
| 13 | PRJNA362897 | P. larvae ssp. *puvifaciens* | ATCC 13537 | Complete | 3 | 4.52 | 44.19 | 4,025 | 24 | 79 | 4 |
| 14 | PRJNA362897 | P. larvae ssp. *puvifaciens* | CCM 38 | Complete | 3 | 4.44 | 44.19 | 4,203 | 24 | 79 | 4 |

_al., 2023_). The outcome of the CD-HIT program is a FASTA format file that contains the entirety of conserved, non-redundant protein sequences, of the _P. larvae_ proteome. In order to identify the essential proteins required for pathogen survival, The selected set of proteins was subjected to BLASTp against the Database of Essential Genes (DEG) (https://tubic.org/deg/public/index.php; (accessed on 11 April 2022)) database (_Zhang & Lin, 2009_) using the standard scoring matrix BLOSUM62, $e$-value = 0.001, and identity 25%. This database serves as a valuable resource in subtractive studies for drug design due to its curated collection of genes (and translated protein products) that are crucial for the survival and basic functions of an organism (_Zhang, Ou & Zhang, 2004_).

The obtained list of proteins was subjected to BLASTp against the human proteome (Homo sapiens, taxid: 9606) and the bee proteome (Anthophila, taxid: 999306) to ensure that any potential therapeutic targets of _P. larvae_ did not share functional similarities with the host proteome (honeybee) or human proteome (end users of honey and its derivatives). Human and bee homologous proteins were eliminated, and only non-homologous proteins were used for further analysis.

Druggability pertains to the capability of biological targets to strongly bind with drugs (_Agoni et al., 2020_). In the context of evaluating the druggability potential of the non-homologous essential protein set of _P. larvae_, we utilized the Drug Bank Database (version 6) (_Knox et al., 2024_), using stand-alone BLAST (_Deng et al., 2007_). Proteins exhibiting non-significant alignments were omitted (similarity < 50% and $E$-value > 0.001), and a refined selection of proteins with promising potential as therapeutic targets was curated (_Basharat, Jahanzaib & Rahman, 2021_). Their effectiveness as therapeutic targets was assessed through an extensive literature review (_Choi et al., 2021_; _López-López et al., 2022_). Additionally, we conducted an evaluation of these _P. larvae_ proteins, encompassing physicochemical attributes (_Artimo et al., 2012_), cellular localization (_Yu et al., 2014_), functional significance, and the prediction of a 3D molecular structure for the selected target.

## 3D structure prediction of the selected therapeutic target

The 3D protein structural prediction analysis was predicted _via_ the AlphaFold (_Ruff & Pappu, 2021_) v2.2 program and PyMol software (_DeLano, 2002_; _Yuan et al., 2016_). AlphaFold, an artificial intelligence software developed by Google's DeepMind, offers accurate predictions of protein structures based on their amino acid sequences. The confidence score, pLDT, provided by AlphaFold, allowed us to assess the quality of protein models, prioritizing regions with pLDT>90 for high-accuracy modeling. To predict the active site of our therapeutic targets, we employed the Computational Atlas of Protein Surface Topography (CASTp) (_Tian et al., 2018_). This tool provides detailed information on surface area, volume, and accessibility of binding sites. It also specifies the amino acids responsible for forming active sites in proteins.

## Virtual screening of phytomolecules against the selected therapeutic target

Virtual screening was attempted using ligands extracted from the Chinese medicine database ($n = 36{,}043$) (_Basharat & Meshal, 2024_). Molecular docking against the selected

therapeutic target (ACP synthase III) was performed using the AutoDockVina software (*Huey, Morris & Forli, 2012*), known for its accurate prediction of binding affinity. Missing hydrogens were added, protein and ligand was convert to pdbqt format and $X, Y$ and $Z$ coordinates of the docking box center were: 43.66, 2.29 and 9.59. Box coordinates were 15.41, 12.17 and 14.39.The screened ligands were ranked by binding energy, with compounds forming stable complexes considered potential enzyme inhibitors. Visualization was done in Maestro v2020-3 (https://www.schrodinger.com/).

### MD simulation study

MD simulation for the top-ranking complex was carried out in order to examine the conformational changes in the protein that resulted from the ligand-binding site and to evaluate the impact of these changes over the protein-ligand complex. The simulation was conducted using the GROMACS (*Abraham et al., 2015*). The parameters were, force field: AMBER ff19SB, water type: TIP3P, ions: NaCl, ligand topology force field: GAFF2, temperature: 298k, pressure: 1 bar, minimization step: 20,000 (on 5 ns). Initial velocity was changed by changing "ntx" and "ig" tontx = 5, ig = 8, from first (C1) to second run (C2) and ntx = 2, ig = 5 during third run (C3). Trajectories were saved and analyzed using CCPTRAJ (*Roe & Cheatham III 2013*).

### Prediction of the toxicity of top hit compounds

The physicochemical parameters related to drug-likeness, adsorption, distribution, metabolism, excretion (ADME) and toxicity were calculated for the top hit compounds using the ADMET prediction servers (http://lmmd.ecust.edu.cn/admetsar2) and SwissADME (http://www.swissadme.ch/). However, only few details were useful as the machine learning models have not yet been trained on honey beeADME data and are only available for human and some model organisms.

## RESULTS

### Pan and core proteome statistics

We examined the pan-proteome (comprising all proteins) of studied strains of *P. larvae* using the BPGA pipeline (Fig. S1A).The core proteins (present in >95% strains), accessory proteins (present in 10% of strains) and unique proteins (strain-specific proteins), provided insights into the shared and distinct genetic elements. COG profile showed maximum number of protein groups for recombination, repair and replication (Fig. S1B). The phylogeny was different based on core and pan proteome profile (Figs. S1C and S1D).

Table 2 depicts a statistical overview of DNA sequences encoding core, unique and accessory proteomes for the analyzed strains. The protein counts reveal significant variation in proteome size, with the total number of proteins ranges from 3,489 to 4,423. Shared or core proteins count was 1,438 for all strains, while the number of accessory proteins varied across strains, with DSM25719 having the highest count at 2,735. Unique proteins ranged from 1 to 293, highlighting the distinctiveness of proteins exclusive to each strain. Overall, BPGA revealed that the pan-proteome was currently open for *P. larvae*, with an expected size of 6,007 and an estimated size of 6,029.06. It means that additional unique

**Table 2  Statistical summary of DNA sequences coding the pan and core proteomes of the analyzed strains.** Missing proteins means that they are not present in the specific strain but are present in other studied strains. (A) 3D interactions of ACP synthase III with top scoring ligands (B) 3D and 2D interaction of ZINC95910054 with active site amino acids of ACP synthase III (C) 3D and 2D interaction of ZINC59586481 with active site amino acids of ACP synthase III (D) 3D and 2D interaction of ZINC14444861 with active site amino acids of ACP synthase III.

| Identifier | Strain names | Total number of proteins | Number of core proteins | Number of accessory proteins | Number of unique proteins | Number of proteins missing |
|---|---|---|---|---|---|---|
| GCA_000153605.1 | BRL-230010 | 3,881 | 1,438 | 2,092 | 29 | 201 |
| GCA_000187665.4 | B-3650 | 4,153 | 1,438 | 1,842 | 25 | 192 |
| GCA_000511115.1 | DSM25719 | 4,375 | 1,438 | 2,735 | 57 | 8 |
| GCA_000511405.1 | DSM-25430 | 3,489 | 1,438 | 2,038 | 58 | 115 |
| GCA_000988145.1 | MEX-14 | 4,015 | 1,438 | 2,314 | 21 | 13 |
| GCA_002003265.1 | ATCC 9545 | 3,864 | 1,438 | 2,143 | 18 | 35 |
| GCA_002007765.1 | ATCC 13537 | 3,864 | 1,438 | 1,952 | 45 | 216 |
| GCA_002043025.1 | CCM 38 | 4,087 | 1,438 | 2,383 | 14 | 8 |
| GCA_002082155.1 | SAG 10367 | 4,259 | 1,438 | 1,974 | 293 | 172 |
| GCA_002951875.1 | RRIC_I | 3,939 | 1,438 | 2,459 | 8 | 2 |
| GCA_002951915.1 | ERIC_III | 4,307 | 1,438 | 2,692 | 96 | 0 |
| GCA_002951935.1 | ERIC_IV | 4,024 | 1,438 | 2,561 | 1 | 0 |
| GCA_011220525.1 | ERIC_ V | 4,423 | 1,438 | 2,630 | 159 | 3 |
| GCA_015912845.1 | G25-75 | 4,147 | 1,438 | 2,171 | 15 | 95 |

proteins are likely to be identified as more organisms are considered. The core proteome, on the other hand, was described as closed, with an expected size of 0 and an estimated size of 1,219.93. This may seem counter intuitive but suggests that these shared proteins are already identified. This implies that further analysis of additional organisms is not expected to contribute new proteins to the core set. The core proteome that supports the fundamental biological traits was used for the subsequent analysis.

## Subtractive proteomics and drug target prioritization

Subtractive proteomics is a novel alternative method for conducting *in silico* studies in order to identify potential therapeutic targets. The main goal of this study is to find a therapeutic target from the *P. larvae* core proteome essential for the pathogen's survival and that has not yet been used for antibiotic therapy. The result of the analysis is summarized in Table 3. The CD-HIT program was used to eliminate eight redundant sequences with an identity cut-off of 60%. The remaining 1,837 non-redundant proteins were subjected to BLASTp against essential proteins present in DEG. The number of non-homologous proteins that is essential for the survival of the pathogen was 187. The removal of 163 hypothetical proteins resulted in a refined set of 24 proteins (Table S1). Hypothetical proteins were excluded from the analysis due to a lack of information about their function and essentiality. This exclusion was essential as the unknown nature of these proteins prevents their use in drug design efforts.

**Table 3  Subtractive genomics for *P. larvae*.** The result of sequences from subtractive proteomics for *P. larvae*.

| Sequences | Number |
| --- | --- |
| Pan-proteome | 62,378 |
| Core proteome | 1,845 |
| Non-redundant sequences (eight sequences removed) | 1,837 |
| Essential proteins available in DEG (elimination of 1,650 sequences) | 187 |
| Hypothetical protein removal (removal of 163 sequences) | 24 |
| Proteins not homologous to bee and human proteome (elimination of 15 sequences) | 9 |
| Druggability analysis (elimination of six sequences) | 3 |
| Literature (elimination of two sequences) | 1 |

Non-homologous analysis was used to find protein targets that are absent from the host (bees) and human to avoid negative effects of the drug on them. Among this set, 15 proteins were discovered to be similar to human proteins and the remaining proteins were selected for inferring druggability. Druggability refers to the capacity of a small-molecule drug to effectively modulate the activity of a therapeutic target with high affinity (*Agoni et al., 2020*). This critical attribute is essential in the identification of potential targets in pathogens. Consequently, the exploration of the ultimate list of potential drug targets within the DrugBank database (*Knox et al., 2024*) led to the identification of three proteins demonstrating  druggable characteristics (Table 4).

Based on a comprehensive literature survey, ketoacyl-ACP synthase III (FabH) emerged as an extensively studied protein employed as a therapeutic target in a range of pathogenic bacterial species due to its involvement in fatty acid biosynthesis, and implications for various cellular processes (*Khandekar, Daines & Lonsdale, 2003*). The strains where it has been studied as a drug target include encompass highly concerning multidrug-resistant strains including *Acinetobacter baumanii* (*Cross et al., 2021*), *Staphylococcus aureus* (*He & Reynolds, 2002*), *Mycobacterium tuberculosis* (*Singh et al., 2011*), *Enterococcus faecium* (*Wang & Ma, 2013*), *Streptococcus pneumonia* (*Khandekar et al., 2001*) *etc.*

### ACP synthase III3D structure determination and active site prediction

The 3D structure of ACP synthase III was predicted employing the Alpha-Fold software, and its depiction is presented in Fig. 1A. The obtained structure achieved a commendable MolProbityScore of 1.78. Additionally, the Ramachandran plot (depicted in Fig. 1B) revealed an impressive 95.20% of residues situated within the highly favored region, with a mere 0.31% falling into the outlier category.

The active site (binding pockets) of the modeled protein was predicted using the CASTp online prediction tool. Approximately 23 binding pockets were predicted for the protein. The binding cavity with the highest volume (1,392.832) was considered in the next stage of our study. Selecting the binding cavity with the highest volume is based on the assumption that a larger cavity can accommodate ligands of various sizes and shapes (*Das et al., 2020*;

**Table 4   List of *P. larvae* druggable proteins.**  List of *P. larvae* proteins with similarities to Drug-Bank proteins.

| Protein name | DEG identifier | Accession | Similarity to bee proteome | Similarity with the Human Proteome | Drugs available in Drug Bank |
|---|---|---|---|---|---|
| Iron-containing alcohol dehydrogenase | DEG10430416 | WP_192807335.1 | No significant similarity | No significant similarity | Carbaphosphonate |
| Ketoacyl-ACP synthase III | DEG10570076 | WP_174567582.1 | No significant similarity | No significant similarity | Cerulenin |
| Isochorismatase | DEG10470205 | WP_024093894.1 | No significant similarity | No significant similarity | Malonyl-CoA Formic acid Isochorismic Acid |

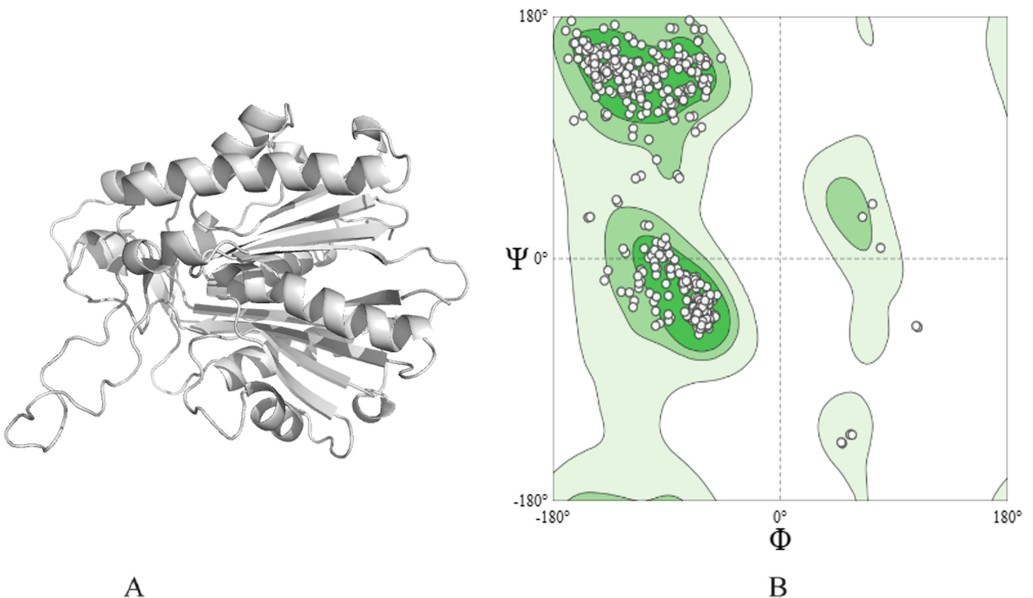

**Figure 1   Protein modeled structure. (A) 3D of ACP synthase III. (B) Ramachandran plot of the structure.**

*Gazgalis et al., 2020*). The amino acids residing in this cavity can play a major role in binding ligand molecules during docking analysis (*Melnikova et al., 2023*).

## Virtual screening for the discovery of ACP synthase III inhibitors

Molecular docking, is a computational technique employed to ascertain the optimal conformation of a ligand within its receptor, based on factors such as position (x, y, z), orientation (qx, qy, qz, qw), and torsion angles (T1, T2…Tn) within the ligand's binding site. Subsequently, the ligands were categorized according to their binding affinities with the target. Among a pool of over 36,000 compounds, 4,267 exhibited interactions with the target. The three ligands demonstrating the lowest binding energies (Fig. S1) are shown with ACP synthase III interactions in Fig. 2.

One of these ligands, ZINC95910054, is derived from celery (*Apiumgraveolens*), a wetland plant belonging to the Apiaceae family, with a longstanding history of cultivation as a culinary vegetable. Comprising a total of 54 atoms, including 26 heavy atoms, its
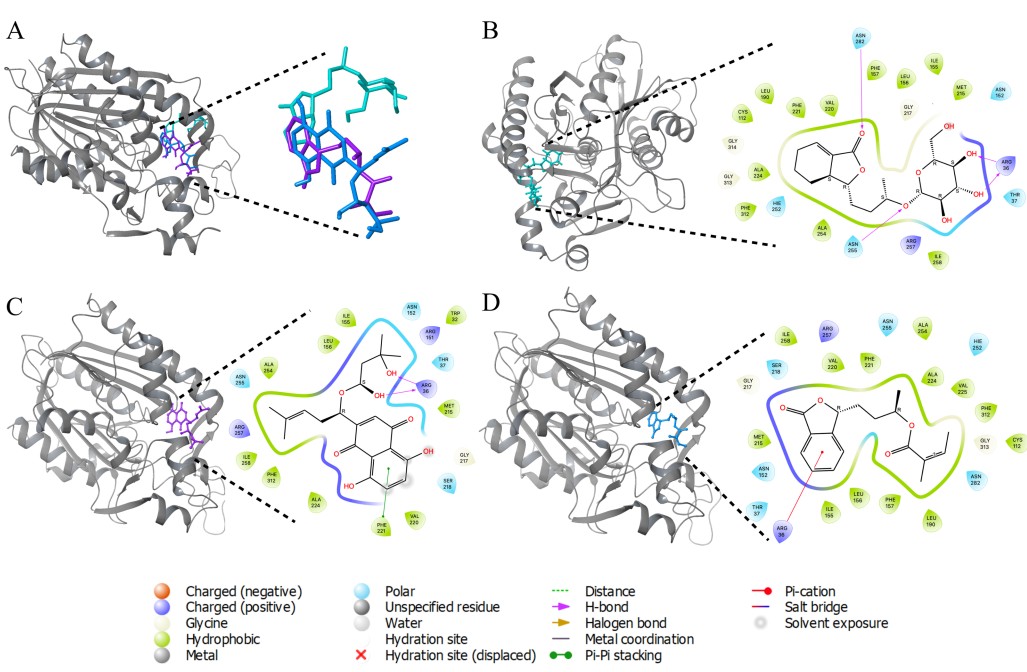

**Figure 2** **3D interactions of ACP synthase III with top scoring ligands.** 3D interactions of ACP synthase III with top scoring ligands (B) 3D and 2D interaction of ZINC95910054 with active site amino acids of ACP synthase III (C) 3D and 2D interaction of ZINC59586481 with active site amino acids of ACP synthase III (D) 3D and 2D interaction of ZINC14444861 with active site amino acids of ACP synthase III.

chemical formula is C18H28O8. Notably, this ligand displayed a binding energy of −10.84 Kcal/mol.

Another ligand, ZINC59586481, demonstrated a docking binding energy of −9.71 Kcal/mol. In contrast, ZINC14444861, originating from the plant *Angelicasinensis* an herbaceous member of the Apiaceae family with a rich medicinal tradition in East Asia exhibited a binding energy of −9.51 Kcal/mol. Comprising a total of 41 atoms, half of which are heavy atoms, this ligand featured 17 carbon atoms in its structure.

## MD simulation

The stability of top ranked docked complexes was inferred by MD simulation studies. The MD simulation study showed a stable system for ZINC95910054 and ACP synthase III, as confirmed by RMSD and RMSF values during 100 ns trajectories (Fig. 3). On the average the RMSD values were less than 3 Å for all runs and RMSF reached upto 7.5 Å around residue 203 (comprising loop-region). Loop regions are more flexible than rest of the protein and depict higher RMSF.

## Toxicity of phytochemicals

The toxicological profiles of the top compounds reveal no toxicity either to honey bees or to humans (Table 5).

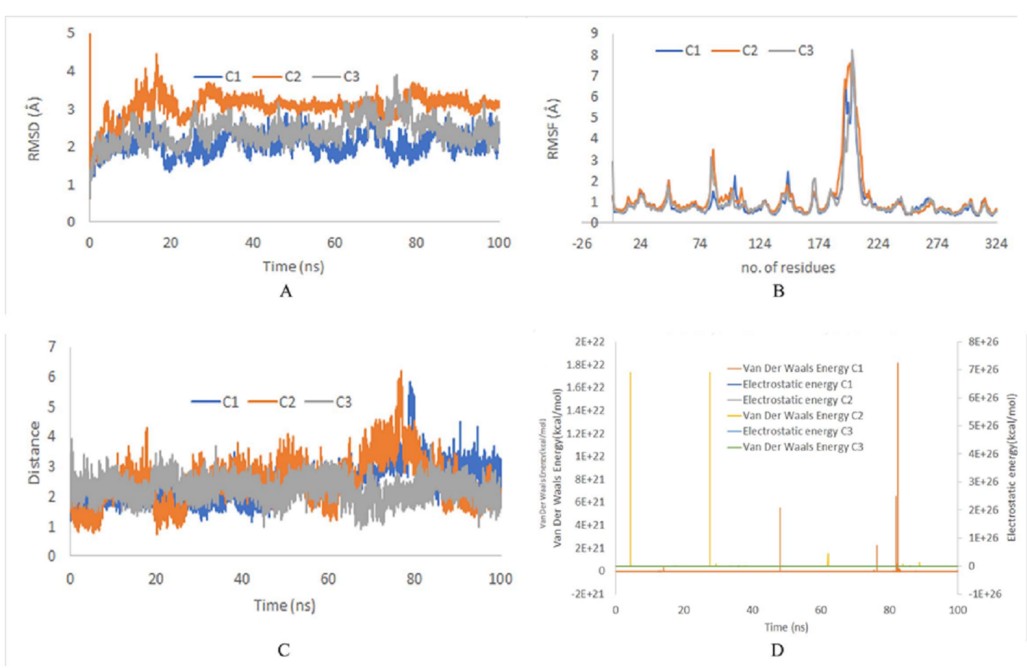

**Figure 3** (A) RMSD of top binding complex ACP synthase III andZINC95910054. C1, C2 and C3 depict the independent runs with different initial velocities. (B) RMSF of top binding ligand ZINC95910054. (C) Distance between the ligand and catalytic site residuesof ACP synthase III.

## DISCUSSION

The worldwide contamination of bee colonies by AFB disease underscores its severity, with the causative agent being the spore-forming bacterium *P. larvae* (*Papić, Diricks & Kušar, 2021*). This contagious disease causes great economic losses in apiculture (*Locke, Low & Forsgren, 2019*). The economic consequences of AFB include reduced honey production and compromised pollination services (*Popovska Stojanov et al., 2021*; *Wakgari & Yigezu, 2021*). Beekeepers may experience financial losses due to the destruction of infected colonies, decreased hive productivity, and the costs associated with disease management measures (*Stanimirović et al., 2019*).

Application and supplementary feeding of antibiotics to restrain clinical AFB symptoms and infection is a common practice in beekeeping (*Masood et al., 2022*; *Zabrodski, 2022*), although it raises concerns about the development of antibiotic-resistant strains of *P. larvae.* The widespread use of antibiotics in beekeeping to control AFB facilitates the dissemination of antibiotic resistance genes and contributes to compromised immunity in honey bees (*Li et al., 2019*). *Daisley et al. (2020)* have reported that administration of oxytetracycline led to an increase in efflux pump resistance (tetB gene abundance) and depletion of crucial symbionts like *Frischella perrera* and *Lactobacillus* strains, known for their roles in regulating immune function and nutrient metabolism. The observed microbial changes manifested in decreased capped brood counts, indicative of compromised hive nutritional status and productivity. Additionally, the antibiotic-induced alterations were

**Table 5  ACP synthase III inhibitors.** Characteristics of three potent ACP synthase III inhibitors.

| Compound | Molecular weight | Molar refraction | Total polar area | Bioavailability score | Toxicity to honey bees |
|---|---|---|---|---|---|
| ZINC95910054 | 372.41 g/mol | 89.93 | 125.68 Å$^2$ | 0.55 | Non toxic score: 0.8103 |
| ZINC59586481 | 390.43 g/mol | 104.14 | 124.29 Å$^2$ | 0.55 | Non toxic score: 0.8542 |
| ZINC14444861 | 288.34 g/mol | 79.84 | 52.60 Å$^2$ | 0.55 | Non toxic score: 0.8637 |

associated with a reduction in the antimicrobial capacity of adult hemolymph, highlighting a decline in immune competence among the honey bees. These findings underscore the intricate relationship between antibiotic exposure, changes in gut microbiota, and the consequential impact on honey bee health and immune functionality (*Ortiz-Alvarado, 2019*). In addition to this, the presence of antibiotic residues in beehive products destined for human consumption are undesirable (*Ye et al., 2023*).

AFB control measures, such as burning infected colonies to prevent the spread of the disease, can result in the loss of equipment and resources invested in the affected hives (*Forsgren et al., 2013*). The economic impact extends beyond individual beekeepers to the broader agricultural sector, as honey bees play a crucial role in pollinating crops, contributing to the overall food production chain. AFB disease control measures also involve relocating adult honey bees to new foundation, eliminating bees and equipment with contamination (*Zabrodski, 2022*), treating colonies with substances like sulphathiazole (*Cervera-Chiner et al., 2020*) or oxytetracycline (*Puvača, 2022*), and sterilizing equipment through methods such as gamma radiation, hot paraffin wax dipping, scorching, or ethylene oxide fumigation (*Kisil & Fotina, 2020*; *Matheson & Reid, 1992*).

Essential oils of several plants have been beneficial in curbing *P. larvae* (*Ansari et al., 2016*; *Puvača, 2022*). Bacteriophages and their lytic enzymes have also emerged as a promising alternative for the treatment and prevention of AFB (*Jończyk-Matysiak et al., 2020*). Probiotic strains have also demonstrated antimicrobial properties against *P. larvae* (*Truong et al., 2023*). Here, we aimed to identify a potent therapeutic target and inhibitor molecules from TCM against the ACP synthase III of *P. larvae*. This was accomplished through the utilization of an integrative computational subtractive proteomics approach, associated with a pan-proteomic assessment of various strains of *P. larvae.*

It resulted in the identification of 1,438 core proteins. Subtractive proteome analysis was also used to identify a collection of 24 proteins non-homologous to humans and honeybees but crucial for the pathogen survival. Ketoacyl-ACP synthase III was chosen as a possible therapeutic target out of the three druggable proteins. The 3D structure of the Ketoacyl-ACP synthase III was modeled and validated and further subjected to molecular docking against TCM compounds. Based on the lowest binding energy, ZINC95910054, ZINC59586481, and ZINC14444861 were prioritized as inhibitors. ZINC95910054 displayed least binding energy of −10.84 Kcal/mol, making it the highest-scoring inhibitor and possibly, the most promising candidate for inhibition of fatty acid synthesis and associated cellular processes in *P. larvae*. Further experimental screening is recommended to identify the most potential ligand among the three.

Different natural strategies based on the application of essential oils, plant extracts, propolis, royal jelly, nonconventional natural molecules, bacteria, and bacteriocines, have been studied *in vitro* and *in vivo* for the prevention and the control of *P. larvae* (*Alonso-Salces et al., 2017*). For instance, *Floris, Carta & Moretti (1996)* tested the antimicrobial activity of 21 types of essential oils against six strains of *P. larvae* using the agar diffusion method. The *in vitro* results showed that the most effective treatment was the cinnamon oil. Furthermore, *Giménez et al. (2021)* found that among thirteen tested natural molecules, menadione, lauric acid, monoglyceride of lauric acid and naringenin showed antimicrobial activity against ten *P. larvae* isolates. This study demonstrated the *in vitro* bactericidal activity of these molecules, specifically menadione and lauric acid, at concentrations practical for field application.

On the other hand, other researchers studied the inhibition of ACP synthase III enzyme in numerous bacterial species. The ACP synthase III (encoded by gene FabH) is one of functional enzymes in fatty acid biosynthesis (FAB), which initiates the FAB cycle by catalyzing the first condensation step between acetyl-CoA and malonyl-ACP (*Zhang, Li & Zhu, 2012*). Thiolactomycin (TLM), a natural compound produced by Actinomycetes has depicted an IC50 value of >100 against ACP synthase of *S. aureus* (*He & Reynolds, 2002*), 110 M against ACP synthase of the *Escherichia coli* (*Price et al., 2001*). Hence, the obtained compound if synthesized and tested *in vivo*, seems promising for the inhibition of *P. larvae* and the control of the AFB.

We acknowledge several limitations of our study. The exclusion of strains with draft or incomplete genomes means that a portion of relevant data was omitted from our analysis. It is also essential to recognize that the strains studied may not fully represent the global *P. larvae* population, given that sequences from various regions worldwide have not been universally sequenced and submitted to NCBI. This potential lack of global diversity in the dataset could impact the overall comprehensiveness of our pan-genome analysis.Future studies incorporating a more diverse range of strains could provide a more comprehensive understanding of the pan-proteome dynamics within this bacterial species. Furthermore, while computational predictions serve as valuable tools, their biological relevance should be experimentally validated to ensure accuracy and applicability. The lack of experimental confirmation may impose limitations on the translational impact of our findings. It is also important to note that computational models may not fully capture unforeseen biological complexities, such as post-translational modifications or protein-protein interactions. Antibiotics used in metaphylaxis are intended to prevent outbreaks of AFB, meaning they are applied in honeybee colonies that do not exhibit any clinical symptoms of the disease. However, the pharmacokinetic profiles of these prioritized compounds are not well-established for honeybees, posing a challenge in predicting their behavior *in vivo*. Furthermore, the absence of comparative data with established antibiotic metaphylaxis makes it difficult to assess the advantages or disadvantages of the prioritized compounds. Despite these constraints, our study serves as a foundational step, paving the way for future research to systematically address these limitations. We recommend further exploration into identifying phytochemicals with similar properties, aiming for a more nuanced inhibition of both *P. larvae* and AFB.

## CONCLUSION

In conclusion, this bioinformatics exploration of essential proteins within *P. larvae* has yielded the identification of three promising candidates as potential therapeutic targets, with potential applications in pharmacology. Notably, ACP synthase III emerged as the primary focus of our research, chosen based on rigorous criteria. Through virtual screening by molecular docking with Autodock software, we scrutinized inhibitor ligands from the traditional Chinese library, ultimately singling out ZINC95910054 as a promising inhibitor for ACP synthase III. Encouragingly, MD simulation affirmed the stability of this complex, bolstering its candidacy as a prospective target for future AFB control. However, the exclusion of certain genotypic information and the reliance on strains with complete genome sequences may influence the generalizability of our findings to the broader *P. larvae* population. Hence, we propose further studies incorporating more diverse strains, compounds and experimental techniques as well.

### Funding

This work was supported by the Ministry of Higher Education and Scientific Research of Tunisia, Grant Number: LR16ES05. The funders had no role in study design, data collection and analysis, decision to publish, or preparation of the manuscript.

### Grant Disclosures

The following grant information was disclosed by the authors:
Ministry of Higher Education and Scientific Research of Tunisia: LR16ES05.

### Competing Interests

Zarrin Basharat is an Academic Editor for PeerJ. Calvin R. Wei is employed by Shing Huei Group.

### Author Contributions

- Sawsen Rebhi conceived and designed the experiments, performed the experiments, prepared figures and/or tables, authored or reviewed drafts of the article, and approved the final draft.
- Zarrin Basharat performed the experiments, analyzed the data, prepared figures and/or tables, authored or reviewed drafts of the article, and approved the final draft.
- Calvin R. Wei performed the experiments, analyzed the data, prepared figures and/or tables, and approved the final draft.
- Salim Lebbal conceived and designed the experiments, authored or reviewed drafts of the article, and approved the final draft.
- Hanen Najjaa analyzed the data, authored or reviewed drafts of the article, and approved the final draft.
- Najla Sadfi-Zouaoui analyzed the data, authored or reviewed drafts of the article, and approved the final draft.

- Abdelmonaem Messaoudi conceived and designed the experiments, analyzed the data, prepared figures and/or tables, authored or reviewed drafts of the article, and approved the final draft.

## Data Availability

The data is available at Zenodo: Rabhi, S., Basharat, Z., Wei, C. R., Sadfi-Zouaoui, N., & Messaoudi, A. (2023). SUPPLEMENTARY (For MD) An integrative pan-genome and subtractive proteomics approach for the identification of potential novel therapeutic drug target against antibiotic resistant honeybee pathogen Paenibacillus larvae [Data set]. Zenodo. https://doi.org/10.5281/zenodo.10115157.

## Supplemental Information

Supplemental information for this article can be found online at http://dx.doi.org/10.7717/peerj.17292#supplemental-information.

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
