# Peer review of "Core proteome mediated subtractive approach for the identification of potential therapeutic drug target against the honeybee pathogen Paenibacillus larvae"

_PeerJ, doi:10.7717/peerj.17292_

## Round 0.1 · original submission · Major Revisions

Please address concerns of all the reviewers and revise manuscript accordingly

In particular, Reviewer 2 has provided extensive additional comments in their PDF.

Reviewer 1 ·

Basic reporting

No comment

Experimental design

Please see attachment

Validity of the findings

Please see attachment.

Annotated reviews are not available for download in order to protect the identity of reviewers who chose to remain anonymous.

Reviewer 2 ·

Basic reporting

First and foremost, the academic writing style of this manuscript should be greatly improved for the international audience to read and understand the manuscript correctly. Multiple vague words or statements are present throughout the body of the manuscript, adding uncertainty to the research methodology and results. Occasional mismatches between material and methods and respective results are present and should be either supplemented with the absent part or eliminated from the manuscript.
More relevant citations need to be added when describing American Foulbrood as a disease.

Experimental design

In the materials and methods section as well as in the results description of the workflow should be improved as it is unclear which categories/values/rules the selection of the protein candidates have been made. In general, a paragraph that discusses the potential limitations of this study should be added to ensure that readers are aware of the study design and its drawbacks appropriately.
Next, the clear distinction between the disease (American Foulbrood) and its causative agent (bacterium P. larvae) should be incorporated throughout the manuscript as it is misleading the reader and indirectly implies that the presence of the bacterium in the hive is a state of the disease, which is not.
In silico analysis also has several logical mismatches, where in materials and methods authors talk about the core genome but in the results discuss a core proteome. Respective corrections should be incorporated into the manuscript to avoid the reader’s confusion.

Validity of the findings

Results of the pan-genome analysis should be discussed and described appropriately as the results reveal the core genome of the tested isolates in this study rather than the P. larvae as a bacterial species. This type of conclusion can be misleading for the reader and should be avoided by incorporating specific details into the presented results.
A section on the literature review of ACP synthase III should include additional phylogenetically relevant examples of the bacteria that have been studied previously and are closely related to the P. larvae and/or share similar cytochemical composition with P. larvae.

Additional comments

Overall, the idea behind the in silico drug discovery use for the screening of potential metaphylactic agents for antibiotic-resistant isolates of P. larvae is promising and should not be neglected. However, this study in particular would have been much scientifically stronger if it would include also at least the in vitro larval rearing of the honey bee larvae challenged with the P. larvae to reveal the true effects of the potential drug.
This manuscript should be thoroughly revised to improve academic writing style, mismatches in the structure of the manuscript and in certain places adjusted for the research design for the publication. In the end by incorporating the appropriate changes and submitting the manuscript for the second round of review should be enough for the manuscript to be accepted for publication.
Please see the detailed revision notes below for further guidance.

Annotated reviews are not available for download in order to protect the identity of reviewers who chose to remain anonymous.

·

Basic reporting

The work is an innovation to detect possible drug targets to stop the pathogenic bacteria P. larvae. Although it is an in silico development, it opens up new pharmacological alternatives and mechanisms to deal with pathogens. This bioinformatics exploration of essential proteins within P. larvae has allowed the identification of three promising candidates as potential therapeutic targets, with potential for applications in pharmacology.

Experimental design

No comment

Validity of the findings

No comment

Additional comments

Many words appear unit you must correct all of them for example:
L16- Correct “Paenibacilluslarvae” for Paenibacillus larvae
L29- Separate “Paenibacilluslarvae” replace from here with P. larvae
L66- Separate “Apismellifera” by Apis mellifera.
These errors must be corrected before publication
This article is promising for publication in the journal.

---

## Round 0.2 · Minor Revisions

Please address the remaining concerns of reviewer #1 and amend the manuscript accordingly.

Reviewer 2 ·

Basic reporting

The revised version of the manuscript titled "Core proteome mediated subtractive approach for the identification of potential therapeutic drug target against the honeybee pathogen Paenibacillus larvae" is a much more improved version of the manuscript that has been originally submitted for peer review. Despite incorporated changes, multiple minor mistakes need to be fixed before the manuscript can be published.

1. Numerous splicing of the words throughout the body of the manuscript. Please adhere to the internationally recognized style of writing academic papers and incorporate a space between words, after commas, and between the dot and the first word of the sentence.
2. Please ensure that throughout the manuscript all names of the living organisms are properly written and italicized.
3. Please re-write the indicated section where confusion regarding American Foulbrood and metaphylactic use of antibiotics is present to avoid biasing the reader's opinion on the application of antibiotics in beekeeping.

Experimental design

Please address minor vague statements throughout the manuscript.
1. Clearly define the threshold for the determination of core, accessory, and cloud proteome.

Validity of the findings

no comment

Annotated reviews are not available for download in order to protect the identity of reviewers who chose to remain anonymous.

·

Basic reporting

No comment

Experimental design

No comment

Validity of the findings

No comment

Additional comments

Pertinent corrections were made to the manuscript.

---

## Round 0.3 · accepted · Accept

All remaining issues were adequately addressed and revised manuscript is acceptable now.